# Environmental Health and Ecological Risk Assessment of Soil Heavy Metal Pollution in the Coastal Cities of Estuarine Bay—A Case Study of Hangzhou Bay, China

**DOI:** 10.3390/toxics8030075

**Published:** 2020-09-22

**Authors:** Rongxi Li, Yuan Yuan, Chengwei Li, Wei Sun, Meng Yang, Xiangrong Wang

**Affiliations:** Department of Environmental Science and Engineering, Fudan University, Shanghai 200438, China; rxli17@fudan.edu.cn (R.L.); yuanyuan17@fudan.edu.cn (Y.Y.); cwli18@fudan.edu.cn (C.L.); wsun16@fudan.edu.cn (W.S.); myang17@fudan.edu.cn (M.Y.)

**Keywords:** risk assessment, environmental health, soil, heavy metals, Hangzhou Bay

## Abstract

Shanghai is the major city on the north shore of Hangzhou Bay, and the administrative regions adjacent to Hangzhou Bay are the Jinshan district, Fengxian district, and Pudong new area (Nanhui district), which are the main intersection areas of manufacturing, transportation, and agriculture in Shanghai. In this paper, we collected a total of 75 topsoil samples from six different functional areas (agricultural areas (19), roadside areas (10), industrial areas (19), residential areas (14), education areas (6), and woodland areas (7)) in these three administrative regions, and the presence of 10 heavy metals (manganese(Mn), zinc(Zn), chromium(Cr), nickel(Ni), lead(Pb), cobalt(Co), cadmium(Cd), mercury(Hg), copper(Cu), and arsenic(As)) was investigated in each sample. The Nemerow pollution index (NPI), pollution load index (PLI), and potential ecological risk index (PERI) were calculated to assess the soil pollution levels. The hazard quotient (HQ) and carcinogenic risk (CR) assessment models were used to assess the human health risks posed by the concentrations of the heavy metals. The CR and HQ for adults and children in different functional areas descended in the following order: industrial areas > roadside areas > woodland areas > residential areas > education areas > agricultural areas. The HQ of Mn for children in industrial areas was higher than 1, and the risk was within the acceptable range.

## 1. Introduction

With the rapid development of urbanization and industrialization worldwide, the quality of urban soil is becoming a serious issue [1,2,3]. The ecological and health risks caused by heavy metal soil pollution from human activities (vehicle emissions, agricultural fertilizer, industrial production activities, and fossil fuel combustion) are receiving more widespread attention worldwide [4,5]. It has become very important for local governments to identify the sources and the degree of risk of heavy metal soil pollution.

Many scholars have researched urban soil pollution around the world in the past decades. For example, Gao et al. (2018) [6] and He et al. (2019) [7] showed that different types of land use have different effects on heavy metal soil concentrations, and Ma et al. (2018) [8] and Han et al. (2018) [9] found that different pollution sources had different effects in different urban functional areas. Agricultural areas are seriously affected by Cr, Hg, and Pb [10,11]; road traffic areas by Zn, Pb, Cu, and Mn [12]; and industrial areas by Cu, Sb, Cd, and Zn [13]. 

The soil conditions in the coastal urban areas of Hangzhou Bay, an estuary bay in China, have attracted attention from scholars [14,15]. However, most studies focused on whole cities (Shanghai, Jiaxing, Hangzhou, Shaoxing, and Ningbo) around Hangzhou Bay, and these cities are readily affected by other geographical factors (e.g., the Yangtze River and Lake Taihu), which could weaken the direct influence of Hangzhou Bay on the soil quality of the main functional area [16]. In addition, most of the research concentrated on the sediments or water animals and plants in Hangzhou Bay, whereas direct research on soil pollution in different functional areas of coastal urban zones is lacking. The main focus of this study was the topsoil in different functional areas of the coastal cities, which are directly affected by Hangzhou Bay and the historical land-use changes of the functional areas where serious pollution was determined. We also identified the main factors causing serious pollution in this area. 

The United States Environmental Protection Agency (USEPA) index system is typically used in research to evaluate the degree of the carcinogenic and non-carcinogenic risks of heavy metal soil pollution, which could affect human health through three exposure pathways (ingestion, inhalation, and dermal exposure) [17]. For the ecological risk assessment in this study, we used a comprehensive method that combined the potential Ecological Risk Assessment, Nemerow Pollution Index, and Pollution Load Index [18,19,20,21]. This method was used to comprehensively and accurately evaluate and grade the pollution degree of various heavy metals in each sample in the study area by combining the heavy metal contents, ecological effects, environmental effects, and biological toxicity according to the properties of heavy metals in the soil and their migration, transformation, deposition, and other behavioral characteristics in the environment.

The coastal city area of Hangzhou Bay is an important intersection area of manufacturing, transportation, and agriculture in the Yangtze River Delta due to the unique geographical location and climate conditions [22]. However, the rapid economic development and human activities have made the soil quality around Hangzhou Bay one of the most polluted areas in China [23]. This study selected three administrative regions (the Jinshan district, Fengxian district, and Pudong new area (Nanhui district) of Shanghai, which are directly affected by the geographical and natural factors of Hangzhou Bay as the research scope. We collected 75 samples from six types of functional areas (agricultural areas (19), roadside areas (10), industrial areas (19), residential areas (14), education areas (6), and woodland areas (7)) in this study area, and 10 heavy metals (Mn, Zn, Cr, Ni, Pb, Co, Cd, Hg, Cu, and As) were detected and analyzed in each sample. 

The primary objectives of this study were to (i) investigate the concentrations of topsoil heavy metals under different land-use types in the study area; (ii) use multivariate statistical methods to identify the main sources and distribution trends of heavy metal pollution; (iii) use the Nemerow pollution index (NPI), pollution load index (PLI), and potential ecological risk index (PERI) to determine the potential ecological risk level of heavy metal pollution; and (iv) assess the carcinogenic and non-carcinogenic risks for adults and children in various functional areas. The overall objective of this study was to understand the actual impact of heavy metal pollution on ecological and health risks in the area and to provide more effective information for local governments to formulate land planning and to effectively prevent soil heavy metal pollution.

## 2. Material and Methods

### 2.1. Study Area and Sample Collection

Shanghai (120°52′–122°12′ E, 30°40′–31°53′ N) (Figure 1) is located in the east of the Yangtze River Delta and is one of the largest cities in China, with an area of 6340.5 km^2^ and a population of approximately 24.83 million people. Shanghai has a subtropical monsoon climate, the average annual temperature is 17.6 °C, the annual sunshine duration is 1885.9 h, and the annual rainfall is 1173.4 mm.

The study area includes three counties in the south of Shanghai (Fengxian, Jinshan, and Pudong), which are located on the north shore of Hangzhou Bay. This area covers 2543 km^2^, and the population is 6.999 million. The proportions of the primary industry, secondary industry, tertiary industry are 12.5%, 55%, and 32.5%, respectively. Thus, the secondary industry, i.e., the leading industry, which includes manufacturing, petrochemical, textile, and electronics industries, could cause serious harm to the environment and human health.

We collected a total of 75 topsoil samples (0–30 cm in depth) from the study area in March 2019. The distribution of the sample sites is shown in Figure 1. The main functional areas and the number of samples were as follows: agricultural areas (19), roadside areas (10), industrial areas (19), residential areas (14), education areas (6), and woodland areas (7). The sampling site coordinates were recorded with a global positioning system (GPS), and the area of each site was 20 m × 20 m. Each soil-sample was a mixture of five sub-samples that were collected at each sampling site (four from the corners and one from the center). The topsoil samples were collected with a plastic spade, stored in PVC (Polyvinyl chloride) packages with labels, and then transferred to the laboratory for further analysis. In the laboratory, plant residues and stones were removed, after which the samples were ground, passed through a 100 mesh (0.15 mm) nylon screen, and then sealed and stored in sampling bottles for subsequent analysis and testing.

### 2.2. Physicochemical Analysis

We detected 10 heavy metals (Mn, Zn, Cr, Ni, Pb, Co, Cd, Hg, Cu, and As) in these samples. First, 0.25 g of sample was placed in a Teflon bomb with an acid mixture (5:4:1 HNO3 + HClO3 + HF) and then heated to 120 °C for 12 h on a heating plate. The acid digestion was repeated until only a negligible amount of white residue remained. Afterwards, the solution was evaporated to dryness and extracted. Following complete digestion, the solution was sieved through a Whatman paper, and the Cu, Cr, Ni, Zn, Pb, Co, Mn, and Cd concentrations in the soil were analyzed by inductively coupled plasma mass spectrometry (ICP-MS, 7900, USA Thermo Fisher). Additionally, the As and Hg concentrations were measured using an atomic fluorescence spectrometer (AFS, AFS 8220, Beijing Titan Instruments, China). According to the method of inductively coupled plasma-mass spectrometry (USEPA 6020B-2014), the detection limits for Cu, Cr, Ni, Zn, Pb, Mn, and Cd were 0.1, 0.1, 0.1, 0.5, 0.1, 0.2, and 0.01 mg/kg, respectively. According to the method of soil quality-analysis of total mercury, arsenic, and lead contents–atomic fluorescence spectrometry (GB/T 22105-2018), the detection limits for As and Hg were 0.01 and 0.002 mg/kg, respectively.

### 2.3. Ecological Risk Assessment Methods of Soil Heavy Metals

#### 2.3.1. Pollution Load Index (PLI)

PLI is primarily used to evaluate each sample point and the overall soil heavy metal pollution degree in a study area [24]. PLI can indicate the most serious heavy metals and the contribution of each heavy metal to environmental pollution [25]. The PLI formula is as follows:(1)$$PI=\frac{C_{i}}{S_{i}},$$
(2)$$PLI=(PI_{1}+PI_{2}+\cdots +PI_{n}{)}^{\frac{1}{n}},$$
where *C*_i_ is the actual concentration of metal *i* in soil (mg kg^−1^), *S_i_* is the evaluation standard of metal *i* (mg kg^−1^), and in this paper, the soil heavy metal background values of Shanghai were used for evaluation standard (Local background values in Table 3), n and m are the number of soil heavy metals and sampling sites, respectively, in this study. The *PLI* classifies four grades: *PLI* < 1, uncontaminated; 1 ≤ *PLI* < 2, uncontaminated to moderately contaminated; 2 ≤ *PLI* < 3, moderately to strongly contaminated; and *PLI* ≥ 3, strongly contaminated.

#### 2.3.2. Nemerow Pollution Index (NPI)

The comprehensive pollution index takes the average value into account and the highest value of the single factor pollution index, which could comprehensively reflect the average pollution level of various pollutants in the soil and highlight the role of more serious pollutants [26,27]. Generally, the Nemerow pollution index (NPI) is used to evaluate the comprehensive pollution of heavy metals in soil [28]. The calculation formula is:(3)$$P_{\mathrm{ave}}=\frac{1}{n}\sum \frac{C_{i}}{S_{i}},$$
(4)$$NPI=\sqrt{\frac{P_{\mathrm{ave}}{}^{2}+P_{\max}{}^{2}}{2}},$$
where *P*_ave_ is the average value of the single pollution index for each heavy metal, and the *P*_max_ is the maximum value of the single pollution index for each heavy metal. The *NPI* is classified as [29]: *NPI* ≤ 0.7, clean; 0.7 < *NPI* ≤ 1, warning limit; 1 < *NPI* ≤ 2, slight pollution; 2 < *NPI* ≤ 3, moderate pollution; and *NPI* > 3, heavy pollution.

#### 2.3.3. Potential Ecological Risk Index (PERI)

The PERI was proposed by Hakanson (1980) [30] to assess the degree of ecological risks of soil heavy metals; this index reflects the toxicity of soil heavy metals and the response of the environment [31]. The PER equation is:(5)$$PERI=\sum_{i}^{n}E_{r}^{i}=\sum_{i}^{n}(T_{n}^{i}\times \frac{C_{i}}{B_{i}}),$$
where *C_i_* and *B_i_* are the measurement concentration and local background value, respectively, of soil heavy metal *i* [25], Ti n is the eco-toxicity response coefficient of heavy metal *i*, and *Ei r* is the *PERI* of heavy metal *i*. The *PERI* classifies five levels: *PERI* < 40, low risk; 40 ≤ *PERI* < 80, moderate risk; 80 ≤ *PERI* < 160, considerable risk; 160 ≤ *PERI* < 320, high risk; *PERI* ≥ 20, very high risk.

### 2.4. Health Risk Assessment Models

Health risks (HR) mainly include non-carcinogenic risk (NCR) and carcinogenic risk (CR). Models of HR are typically used to quantify the human health risks of soil heavy metals from three main pathways (ingestion, inhalation, and dermal exposure) based on the USEPA protocol [32,33,34,35,36]. In this study, these models were used to evaluate the impact of heavy metal pollution on local adults and children. The equations of these models are as follows and the abbreviations are defined in Table 1 and Table 2.

Adults:(6)$$\begin{array}{l} CDI_{a-ing}=\frac{C_{i}\times IRg_{adult}\times CF\times EF\times ED_{adult}}{BW_{adult}\times AT}\\ CDI_{a-inh}=\frac{C_{i}\times IRh_{adult}\times CF\times EF\times ED_{adult}\times FSPO\times PLAF\times PM_{10}}{BW_{adult}\times AT}\\ CDI_{a-derm}=\frac{C_{i}\times CF\times SA_{adult}\times AF\times ABS\times EF\times ED_{adult}}{BW_{adult}\times AT} \end{array},$$

Children:(7)$$\begin{array}{l} CDI_{c-ing}=\frac{c\times CF\times EF}{AT}\times (\frac{IRg_{adult}ED_{adult}}{BW_{adult}}+\frac{IRg_{children}ED_{children}}{BW_{children}})\\ CDI_{c-inh}=\frac{c\times CF\times EF\times FSPO\times PLAF\times PM_{10}}{AT}\times (\frac{IRh_{adult}ED_{adult}}{BW_{adult}}+\frac{IRh_{children}ED_{children}}{BW_{children}})\\ CDI_{c-derm}=\frac{c\times CF\times AF\times ABS\times EF}{AT}\times (\frac{SA_{adult}ED_{adult}}{BW_{adult}}+\frac{SA_{children}ED_{children}}{BW_{children}}) \end{array},$$
(8)$$HQ=\sum \frac{CDI}{RfD}.$$
CR=CDI×SF

### 2.5. Statistical Analysis Methods

All spatial distribution maps of topsoil heavy metals were obtained using geographic information system software (ArcGIS 10.2). Principal component analysis (PCA) was primarily used to classify the soil heavy metals that may have the same source and to identify their potential sources [37,38]. The PCA, PLI, NPI, PERI, HQ, and CR were performed using SPSS 23.0 software and Microsoft Excel 2007.

## 3. Result and Discussion

### 3.1. The Concentration and Spatial Distribution of Heavy Metals

The concentrations of the 10 topsoil heavy metals in the six functional areas are shown in Table 1. Mn had the highest mean value (837.3 mg/kg), followed by Zn (98.3 mg/kg), Cr (96.9 mg/kg), Ni (39.3 mg/kg), Cu (28.4 mg/kg), Pb (27.5 mg/kg), Co (14.7 mg/kg), As (9.3 mg/kg), Cd (0.18 mg/kg), and Hg (0.11 mg/kg), and they are between 0.8 and 2 times the background value of soil in China. The coefficient of variation (CV) represents the average dispersion degree of soil heavy metal concentrations, with CV < 15%, reflecting low variation; 15% < CV ≤ 35%, reflecting moderate variation; and CV > 35% reflecting high variation (Xu et al., 2014). The CV of these metals in the whole area descended in order as follows: Hg (60.8%) > Cd (30.0%) > Mn (26.4%) > Pb (26.0%) > Cu (22.4%) > Zn (22.2%) > As (20.2%) > Ni (16.4%) > Cr (13.2%) > Co (10.4%).

According to the analysis, the CVs of these metals indicated that Hg had high variation and the concentration varied greatly in different areas: Cd, Mn, Pb, Cu, Zn, As, and Ni demonstrated moderate variation, and the concentration distributions were relatively uneven; Cr and Co demonstrated low variation, and the concentration distributions were relatively uniform. The order of CVs in the six functional areas was the same as that for the whole area, which revealed that human activities had significant interference effects on the heavy metal enrichment in the whole area except for Co and Cr (Table 3).

The spatial distribution trends of soil heavy metals in the study area are shown in Figure 2. Mn and total heavy metals showed similar trends. We examined the relationship between each heavy metal and found that Mn had the strongest correlation with total heavy metals, which indicated that Mn might be a good indicator of soil heavy metal concentration in this area [39]. Higher concentrations of Hg and Ni were found in the east and they had similar distribution trends, indicating that Hg and Ni may have the same source associated with petrochemical industry [40].

The CVs of Co and Cr were low, and the distribution was relatively uniform, which revealed that Co and Cr might be from a natural source. The concentrations hot-spots of Pb were mainly found in the south and north east of the study area, and they had some common distribution characteristics with Zn, indicating that Pb and Zn were associated with busy local traffic and the airport [28,41]. As, Cd, and Cu had the same distribution pattern, which indicated that they may have the same source associated with the local heavy equipment manufacturing industry [13,42].

Table 4 shows the number of sample points where the concentration of heavy metals was more than twice the Chinese background value. Cd and Hg had obvious enrichment points in all functional areas. As the main enrichment of Cd and Hg in the soil resulted from road traffic and industrial activities, we investigated the main enrichment points of the other functional areas. In the agricultural and education areas, the land-use of these sample points (10, 12, 20, 21, 33, 43, 47, 55, 64, and 67), which had serious Cd and Hg heavy metal pollution, had not changed in the past decade.

The primary pollutants came from the surrounding road traffic and industrial park pollution discharge. The sample points in residential areas (18, 32, 41, 60, and 65) were mostly located downtown with heavy traffic and were seriously affected by automobile exhaust emissions. The sample points (16, 29, and 73) in woodland areas were affected by the industrial point source surrounding them, and sampling point 9 was influenced by the historical problems of heavy metal accumulation in the soil due to the change of land-use from an industrial area to a woodland area. 

The results showed that road traffic and industrial production had obvious enrichment effects on the heavy metal concentrations in the soil of all functional areas, particularly in residential areas, which could cause direct harm to the health of local residents. In addition, the land-use in certain serious soil pollution areas was changed by the government in order to reduce the local heavy metal concentration; however, the effects were not apparent in the short term.

### 3.2. Statistical Analysis

PCA was used for the agricultural area, roadside area, industrial area, residential area, education area, woodland area, and all area data. After data analysis, the roadside areas, education areas, and woodland areas were not suitable for PCA, as the sample numbers of these three functional areas were less than 10. The data of the agricultural areas, industrial areas, residential areas, and all areas conformed to the positive definite matrix, which could use PCA (varimax rotation mode) to identify the associated PCs (Table 5).

In the industrial areas, PCA extracted three factors, which explained 80.664% of the total variance of the data. PC1a explained 33.104% of the total variance. Cu, Pb, Cd, and As showed strong positive loadings of 94.5%, 87.7%, 66.5%, and 87.2%, respectively. These results indicated that Cu, Pb, Cd, and As could be identified as anthropogenic and might be from sources similar to the transportation industry, chemical industry, and fossil fuel combustion. PC2a explained 25.781% of the total variance, and the strong positive loadings of Cr, Ni, Mn, and Co were 89.5%, 91.8%, 47.6%, and 69.1%, respectively. PC3a explained 21.779% of the total variance and showed strong positive loadings of Zn and Hg.

In the agricultural area, PCA extracted three factors, which explained 78.627% of the total variance of the data. PC1b explained 47.643% of the total variance and contained Cu, Cr, Ni, Zn, Pb, Mn, and Co, with loading values of 87.1%, 71.9%, 85.5%, 86%, 89.3%, 74.8%, and 77.4%, respectively. PC2b explained 15.896% of the total variance, and Cd showed a strong negative loading of −93.9%. PC3b explained 15.089% of the total variance, and the loading values of As and Hg were 76.9 and 76.5%, respectively.

In the residential areas, two factors were extracted by PCA, which explained 78.776% of the total variance of the data. PC1c explained 41.606% of the total variance, and Cu, Cr, Zn, Pb, Cd, As, and Hg showed strong positive loadings. PC2c explained 36.170% of the total variance and the loading values of Ni, Mn, and Co were 95.8%, 91.6%, and 94.2%, respectively.

In all areas, two factors were extracted, which explained 63.939% of the total variance of the data. Cu, Zn, Pb, Cd, As, and Hg had strong positive loadings of 81.7%, 74.7%, 72.5%, 84.5%, 66.5%, and 66.8% on PC1, which explained 35.711% of the total variance. PC2 explained 28.228% of the total variance of Cr, Ni, Mn, and Co.

The results of PC1a, PC1b, and PC1c indicated that Cu and Pb had certain homologies and were mainly from automobile exhaust emissions, fossil fuel combustion, and heavy metal smelting [5]. However, there were some differences among them, which indicated that there were different proportions of major pollution sources in these three functional areas (industrial areas, agricultural areas, and residential areas). The PCAc results were similar to those of PCAd, indicating that the soil heavy metal pollution sources in residential areas were the same as those in the whole area, which is different from the result presented by Hu et al. (2014) [43].

### 3.3. Ecological Risk Assessment (ERA)

In this study, PLI, NPI, and PERI were used to evaluate the ecological risk levels of soil heavy metal concentrations in the study area, and the Environmental Quality Standard for Soils (GB 15618-1995) was used for the background value of the soil heavy metal concentrations. Since Cr is not toxic in soil, so Cr was not considered in the ecological risk assessment of this paper. The PLI ranges (Figure 3) in different functional areas were 1.04 to 1.76 in industrial areas (mean 1.34), 1.12 to 1.48 in agricultural areas (mean 1.29), 0.95 to 1.60 in residential areas (mean 1.27), 1.16 to 2.02 in roadside areas (mean 1.35), 1.08 to 1.68 in woodland areas (mean 1.35), 0.97 to 1.56 in education areas (mean 1.22), and 0.95 to 2.02 in all areas (mean 1.30). The PI of 9 heavy metals in the industrial area descended in order as follows: Hg > Cd > Mn > Ni > Zn > Cu > Pb > Co > As, in agricultural areas: Hg > Cd > Ni > Mn > Zn > Cu > Co > Pb > As; in residential areas: Hg > Cd > Mn > Ni > Zn > Cu > Co > Pb > As; in roadside areas: Cd > Mn > Zn > Ni > Hg > Cu > Co > Pb > As; in woodland areas: Cd > Hg > Ni > Mn > Zn > Cu > Co > Pb > As; in education areas: Cd > Mn > Ni > Hg > Zn > Cu > Co > Pb > As; and in all areas: Cd > Hg > Ni > Mn > Zn > Cu > Co > Pb > As. The sequence of PI in woodland areas was the same as in all areas. In the six functional areas, the PLI value of As in all areas and the PLI value of Pb in agricultural areas and education areas were less than 1, indicating uncontaminated status; the PLI value of Cd in woodland areas was 2.03, indicating moderately to strongly contaminated; the PLI values of heavy metals in other areas ranged from 1 to 2, indicating uncontaminated to moderately contaminated. In addition, the PLI value of Cd was the highest in residential areas, roadside areas, woodland areas, education areas, and was the second highest in industrial areas and agricultural areas. This result indicated that the pollution level of Cd in all functional areas was serious and widespread.

The NPI values of the soil heavy metals in the six functional areas were all greater than 1, except for As in the woodland areas and education areas, indicating that different functional areas had enrichment effects on the 9 soil heavy metals (Figure 4). The NPI value of Hg in industrial areas was the highest and was much higher than that in other functional areas (there was light contamination in the education areas, moderate contamination in the roadside areas, and heavy contamination in others), indicating that the enrichment of Hg was the most serious. In addition, the NPIs of Cu, As, and Co indicated light contamination (1 < NPI < 2) in the six functional areas; the NPLs of Cr, Ni, Zn, Pb, and Mn indicated moderate contamination (2 < NPI < 3) in the industrial areas and light contamination in others; the NPL of Cd indicated heavy contamination (NPI > 3) in the roadside areas and moderate contamination in others. The NPL values of soil heavy metals in the study area descended in order as follows: Hg > Cd > Mn > Zn > Ni > Cu > Pb > Co > As. The NPI values of Hg and Mn were relatively high, indicating that the maximum and minimum values of the concentrations of these two heavy metals differed greatly in different regions and were most seriously affected by human interference.

The single risk index and potential ecological risk index of soil heavy metals in the six functional areas are shown in Figure 5. The PERI value of soil heavy metals in different functional areas (except Cd in the roadside area, which was higher than Hg) descended in order as follows: Hg > Cd > As > Ni > Cu > Co > Pb > Mn > Zn. The PERIs of Cd and Hg indicated a moderate risk (40 ≤ PERI < 80), whereas other heavy metals had a low risk (PERI < 40). The toxic response factors of Cd and Hg were much higher than those of the other eight heavy metals, indicating that Cd and Hg were more harmful to human health than other metals. In addition, the PERI value of Cd in the woodland area was highest, which was mainly influenced by point sources, such as electroplating and nonferrous metal smelting factories around this region; the PERI value of Hg in the agricultural areas was the highest, and this was mainly affected by non-point sources, such as atmospheric subsidence, organic fertilizer, and sewage irrigation. The RI value indicated that there were moderate potential risks in all six functional areas. In descending order, the RI values for the six functional areas were woodland areas > agricultural areas > industrial areas > residential areas > roadside areas > education areas. This result indicated that the point source pollution in the study area was greater than the non-point source pollution, especially around the woodland areas.

### 3.4. Health Risk Assessment (HRA)

We used the non-carcinogenic and carcinogenic health risk assessment models from USEPA to analyze the hazard degrees of heavy metals in different functional areas. HQ, and CR were calculated for adults and children based on the three main pathways (ingestion, inhalation, and dermal exposure) (Figure 6).

#### 3.4.1. Non-Carcinogenic Health Risk Assessment

The non-carcinogenic risks of the 10 soil heavy metals in different functional areas were shown to have the same order: Mn > Cr > As > Co > Pb > Ni > Cu > Hg > Zn > Cd for adults and Mn > Cr > As > Pb > Co > Ni > Cu > Hg > Zn > Cd for children. Mn was the major non-carcinogenic heavy metal for humans, and Mn had certain significant non-carcinogenic risks in industrial areas. The non-carcinogenic risk exposure degree of the three exposure pathways in all functional areas were ingestion > dermal exposure > inhalation.

The exposure dose of ingestion was 98.8% of the total dose, which was the major exposure pathway of non-carcinogenic risk from soil heavy metals in the study area. The order of the total non-carcinogenic risk in different functional areas was industrial areas > roadside areas > woodland areas > residential areas > education areas > agricultural areas for children and industrial areas > roadside areas > woodland areas > education areas > residential areas > agricultural areas for adults. In addition, for adults, the hazard index value of heavy metals in each functional area was less than 1, indicating that the non-carcinogenic risk was acceptable; however, for children, the hazard index value of heavy metals in all functional areas was greater than 2, indicating that the soil heavy metals in each functional area had non-carcinogenic risks for children. Therefore, the recommendation is to keep children’s hands and mouths clean to avoid soil heavy metal intake when they play outside.

#### 3.4.2. Carcinogenic Health Risk Assessment

The carcinogenic risk of Co, As, Ni, and Cd in order was Co > Ni > As > Cd. The average values of Co, Ni, As, and Cd were less than 10^−6^, indicating that these four heavy metals showed no significant carcinogenic risk in the study area. The order of the carcinogenic risk in the six functional areas was industrial area > roadside area > woodland area > residential area > education area > agricultural area for adults and children, which is consistent with the order of the non-carcinogenic risk. The carcinogenic risk in the industrial areas was highest but was within the acceptable range.

## 4. Conclusions

We determined the concentrations of 10 topsoil heavy metals (Cu, Cr, Ni, Zn, Pb, Cd, As, Hg, Mn, and Co) in six functional areas (agricultural areas, roadside areas, industrial areas, residential areas, education areas, and woodland areas). The three administrative regions (Jinshan district, Fengxian district, and Pudong new area (Nanhui district)) of Shanghai are on the north shore of Hangzhou Bay. We used the PLI, NPI, and PERI for ecological risk assessment, and non-carcinogenic and carcinogenic risk assessment models were used for human health risk assessment.

(1)The concentrations of Cu, Cr, Ni, Zn, Pb, Cd, Hg, Mn, and Co exceeded the background values of Chinese soil. Among these heavy metals, Hg exhibited the highest CV value (high variation). The overall concentration distribution trend of these heavy metals (except As) was higher in the west, lower in the middle, and intermediate in the east. Many chemical industry parks located in the Jinshan district and the heavy equipment manufacturing and logistics industries located in the Pudong new area (Nanhui district) contribute largely to the heavy metal contamination in the west and east of the study area.(2)The PCA results showed that the pollution sources of soil heavy metals demonstrated differences in these six functional areas. The production of chemical products and the burning of fossil fuels are the primary pollution sources in the industrial areas. Automobile exhaust emissions, atmospheric substances, and the use of organic fertilizers are the main pollution sources in the agricultural and residential areas. The main pollution sources of the woodland areas are the surrounding factories, such as electroplating factories and foundry factories.(3)The cross-contamination between different functional areas was strong. In particular, the contamination in industrial areas and roadside areas had direct impacts on other functional areas in the whole area.(4)The analysis of the three ecological risk assessment models showed that the potential ecological risk of woodland areas was higher than that of agricultural and industrial areas. The impact of industrial pollution sources (mainly rubber plants, power stations, foundries, and chemical plants) on soil quality was highest. These were the main pollution sources for Hg and Cd, which had the greatest ecological risks in this study.(5)The impacts of soil heavy metal pollution in the six different functional areas were low for adults; however, there were certain non-carcinogenic risks for children. Co, As, Ni, and Cd showed no significant carcinogenic risk, but this result serves as a warning for local humans. Due to the many industrial areas in the west and east of the study areas, the soil pollution was more serious. Therefore, it is necessary to formulate policies in these two regions to reduce the level of soil pollution and improve the level of ecological security.

## Figures and Tables

**Figure 1 toxics-08-00075-f001:**
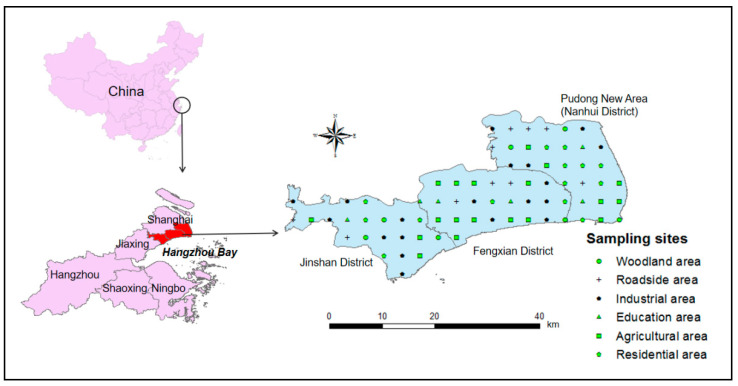
A map of the soil sampling sites and study area in Shanghai.

**Figure 2 toxics-08-00075-f002:**
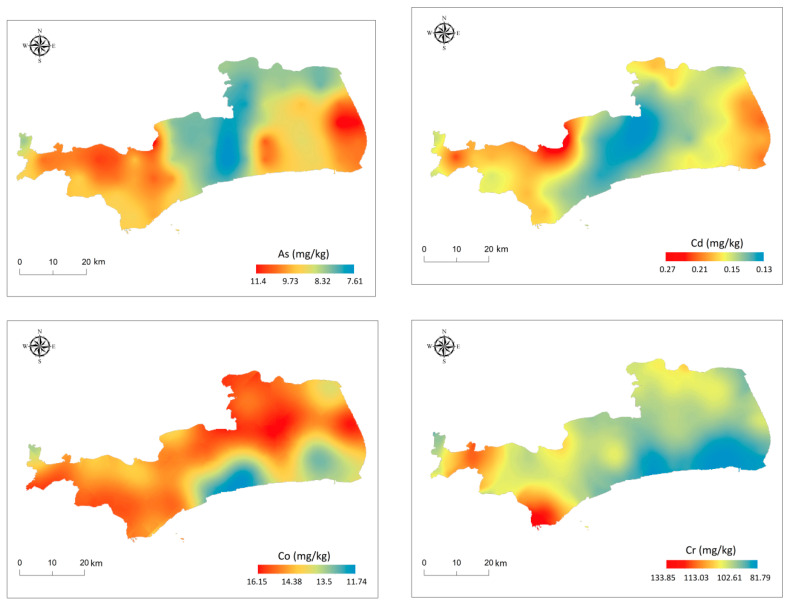
Spatial distribution maps of topsoil heavy metals in the study area.

**Figure 3 toxics-08-00075-f003:**
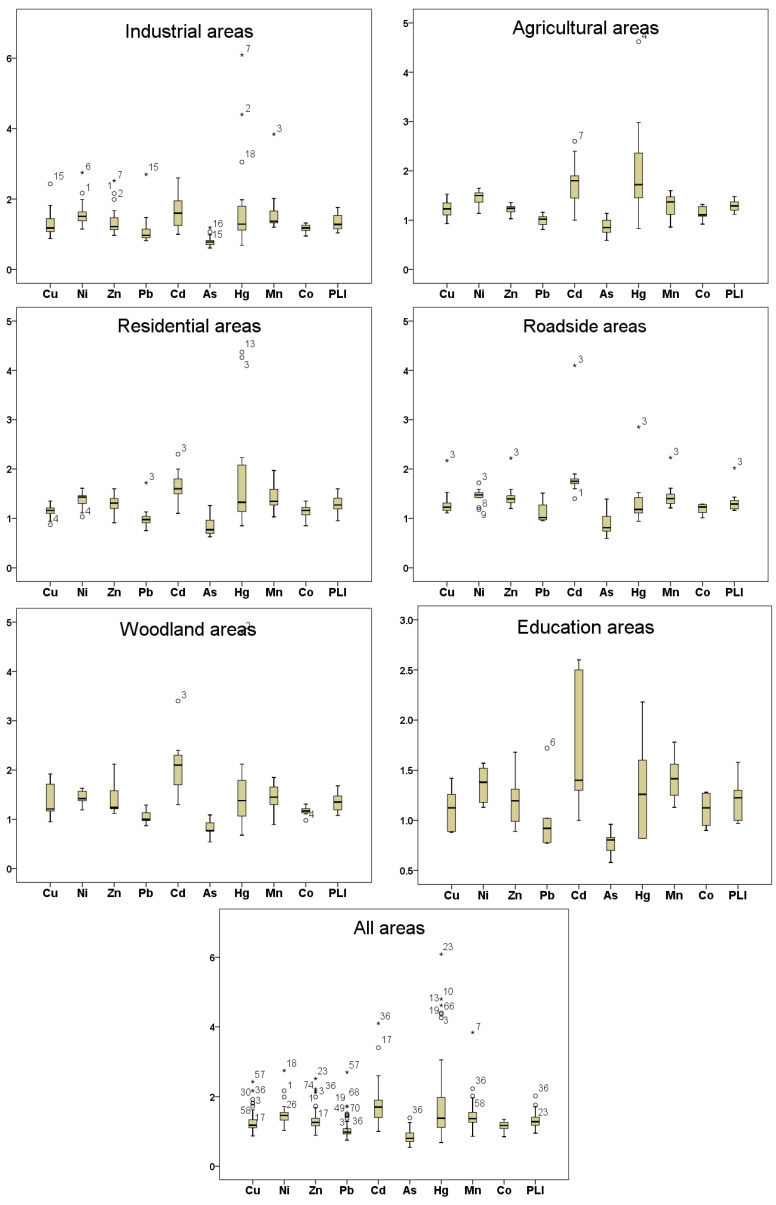
The box-plots of the pollution load index (PLI) values for 9 heavy metals in the six functional areas and all areas. (*x*-axes: the PLI values, *y*-axes: the sample point number of each heavy metal, *: extreme value, discrete value).

**Figure 4 toxics-08-00075-f004:**
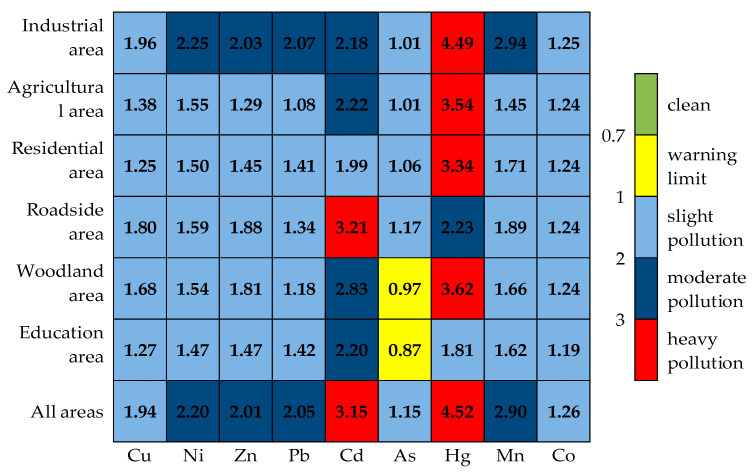
The Nemerow pollution index (NPI) values for the 9 heavy metals in the six functional areas and all areas.

**Figure 5 toxics-08-00075-f005:**
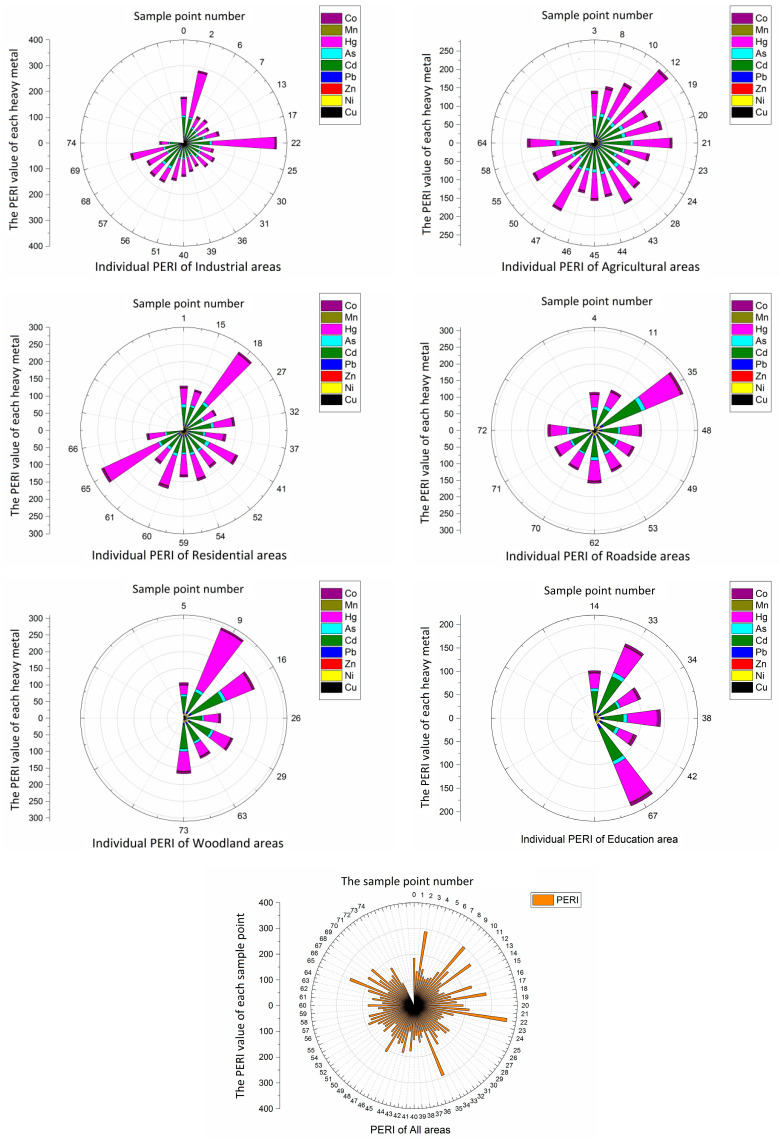
The wind roses of the potential ecological risk index (PERI) values for 9 heavy metals in the six functional areas and all areas.

**Figure 6 toxics-08-00075-f006:**
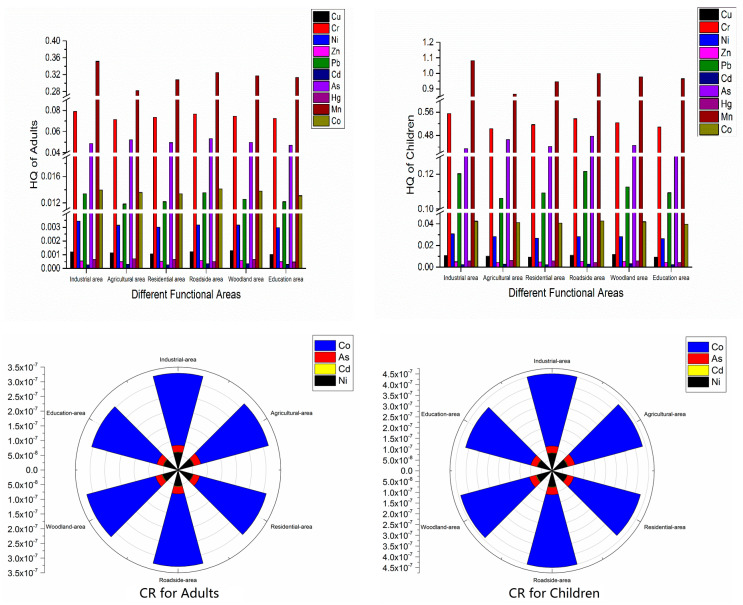
Column charts of the non-carcinogenic risk and wind roses of the carcinogenic risk for adults and children in the six functional areas.

**Table 1 toxics-08-00075-t001:** Meaning of parameters in carcinogenic and non-carcinogenic risk models.

Parameter	Meaning	Value		Unit
		Adult	Children	
CDI	Average chemical daily intakes of human through 3 exposure pathways	−	−	-
C_i_	Measurement concentration of soil heavy metal i	−	−	mg/kg
IRg	Soil intake frequency of human for one day	100	200	mg/d
IRh	Breathing rate of human for one day	14.5	7.5	m^3^/d
CF	Conversion factor	1 × 10^−6^	1 × 10^−6^	kg/mg
EF	Exposure frequency for one year	350	350	d/y
ED	Average exposure duration	24	6	y
BW	Average body weight	60	15	kg
AT	Average time (non-carcinogenic)	ED × 365	ED × 365	d
	Average time (carcinogenic)	70 × 365	70 × 365	
FSPO	Proportion of particulate matter from soil in the air	0.15	0.15	−
PLAF	Inhaled retention rate of particulate matter from soil	0.75	0.75	−
PM_10_	Amount of inhalable particles	0.15	0.15	mg/m
SA	Exposed surface area of skin	4350	1600	cm^2^
AF	Dermal adherence factor	0.07	0.2	mg/(cm^2^ × d)
ABS	Dermal absorption factor	0.001	0.001	−
RfD	Reference doses	−	−	mg/(kg × d)
SF	Slope factors	−	−	−
HQ	Hazard quotient	−	−	−
CR	Carcinogenic risk	−	−	−

**Table 2 toxics-08-00075-t002:** Heavy metal risk reference slope factors (SFs) and reference doses (RfDs; mg/(kg·d)).

Metal	RfD_ing_	RfD_derm_	RfD_inh_	SF_derm_
Cu	4 × 10^−2^	4 × 10^−2^	1.2 × 10^−2^	
Cr	3 × 10^−3^	2.86 × 10^−5^	6 × 10^−5^	
Ni	2 × 10^−2^	2.06 × 10^−2^	5.4 × 10^−3^	8.4 × 10^−1^
Zn	0.3	0.3	0.06	
Pb	3.5 × 10^−3^	3.52 × 10^−3^	5.25 × 10^−3^	
Cd	1 × 10^−3^	1 × 10^−3^	5 × 10^−5^	6.4
As	3 × 10^−4^	3.01 × 10^−4^	1.23 × 10^−4^	1.5
Hg	3 × 10^−4^	8.57 × 10^−5^	2.1 × 10^−5^	
Mn	4.6 × 10^−2^	1.43 × 10^−5^	1.84 × 10^−3^	
Co	2 × 10^−2^	5.71 × 10^−6^	1.60 × 10^−2^	9.8

**Table 3 toxics-08-00075-t003:** The concentrations of 10 topsoil heavy metals (mg/kg) in the six functional areas.

Sampling Site		Cu	Cr	Ni	Zn	Pb	Cd	As	Hg	Mn	Co
	Detection Limit	0.1	0.1	0.1	0.5	0.1	0.01	0.01	0.002	0.5	0.5
Industrial areas (n = 19)	Mean	29.7	102.3	42.6	102.8	29.2	0.16	9.03	0.120	931.4	15.0
	Minimum	19.8	79.5	30.9	72.3	21.3	0.10	6.86	0.045	698.0	12.1
	Maximum	55.0	153.0	74.0	187.0	70.2	0.26	13.30	0.396	2240.0	16.8
	CV (%)	28.6	19.0	23.6	30.4	38.4	30.8	18.1	75.3	36.8	9.2
Agricultural areas (n = 19)	Mean	27.6	92.7	38.8	90.6	25.8	0.18	9.68	0.130	747.1	14.6
	Minimum	21.1	83.0	30.8	76.1	21.0	0.10	6.56	0.054	502.0	11.7
	Maximum	34.5	102.0	44.4	101.0	30.2	0.26	12.80	0.300	932.0	16.8
	CV (%)	13.9	7.2	10.3	6.9	9.4	21.7	18.8	45.4	17.9	10.6
Residential areas (n = 14)	Mean	25.9	95.3	37.0	95.0	26.6	0.16	9.19	0.12	815.6	14.4
	Minimum	19.7	76.9	27.6	67.5	19.6	0.11	7.00	0.055	598.0	10.8
	Maximum	30.5	110.0	43.2	119.0	44.7	0.23	14.10	0.284	1150.0	17.1
	CV (%)	11.6	10.3	11.9	15.6	22.1	18.8	22.0	62.5	17.9	12.1
Roadside areas (n = 10)	Mean	30.2	99.08	39.1	108.2	29.5	0.20	9.91	0.089	861.2	15.1
	Minimum	25.0	85.1	31.7	89.3	24.6	0.14	6.73	0.061	703.0	12.8
	Maximum	49.1	120.0	46.3	165.0	39.3	0.41	15.60	0.185	1300.0	16.4
	CV (%)	23.6	10.5	11.0	20.0	18.9	39.4	25.70	40.0	19.5	8.6
Woodland areas (n = 7)	Mean	31.8	96.6	38.9	106.6	27.4	0.21	9.25	0.120	841.1	14.8
	Minimum	21.4	81.3	31.9	83.2	22.5	0.13	6.08	0.044	520.0	12.5
	Maximum	43.4	108.0	43.8	157.0	33.6	0.34	12.20	0.312	1080.0	16.6
	CV (%)	26.5	11.1	10.6	25.1	14.9	32.5	21.5	77.9	22.8	8.9
Education areas (n = 6)	Mean	25.2	93.9	36.5	90.0	26.6	0.17	8.73	0.090	831.8	14.1
	Minimum	19.8	85.0	30.3	65.7	19.9	0.10	6.55	0.053	657.0	11.4
	Maximum	32.0	113.0	42.3	125.0	44.8	0.26	10.70	0.142	1040.0	16.2
	CV (%)	18.9	11.8	13.6	23.3	34.9	39.7	16.2	40.7	16.3	14.6
All areas (n = 75)	Mean	28.4	96.9	39.3	98.3	27.5	0.18	9.34	0.110	837.3	14.7
	Minimum	19.7	76.9	27.6	65.7	19.6	0.10	6.08	0.044	502.0	10.8
	Maximum	55.0	153.0	74.0	187.0	70.2	0.41	15.60	0.396	2240.0	17.1
	CV (%)	22.4	13.2	16.4	22.2	26.0	30	20.2	60.8	26.4	10.4
Background values of China ^a^		22.6	61.0	26.9	74.2	26.0	0.10	11.20	0.065	583.0	12.7
Local background values ^b^		28.6	75.0	40.0	86.1	25.5	0.13	9.11	0.102	555.5	12.7

^a^ The background values of the heavy metals in the table were obtained from Soil Environmental Quality Standard (GB15618-1995). ^b^ The local background values of the heavy metals in the table were obtained from National Background Values of Soil Elements (1990).

**Table 4 toxics-08-00075-t004:** The number of sample points with serious heavy metal pollution in the six functional areas.

Functional Areas	Sample Point No.									
	Cu	Cr	Ni	Zn	Pb	Cd	As	Hg	Mn	Co
Industrial areas	56	0, 17, 25	0, 17	0, 22	56	0, 2, 22, 56, 57	−	2, 22, 69	6, 57	−
Agricultural areas	−	−	−	−	−	21, 43, 44, 64	−	10, 12, 20, 21, 43, 47, 55	−	−
Residential areas	−	−	−	−	−	18, 32	−	18, 41, 60, 65	−	−
Roadside areas	35	−	−	35	−	35	−	35	35	−
Woodland areas	−	−	−	73	−	9, 16, 29, 73	−	9, 16	−	−
Education areas	−	−	−	−	−	33, 67	−	67	−	−

**Table 5 toxics-08-00075-t005:** Principal components (PC) of industrial areas, agricultural areas, residential areas, and all areas.

Element	Industrial Areas			Agricultural Areas			Residential Areas		All Areas	
	PC1a	PC2a	PC3a	PC1b	PC2b	PC3b	PC1c	PC2c	PC1	PC2
Cu	0.945	0.192	0.189	0.871	−0.216	0.271	0.819	0.351	0.817	0.331
Cr	0.080	0.895	0.262	0.719	0.228	0.084	0.691	0.542	0.380	0.712
Ni	0.043	0.918	0.085	0.855	0.301	−0.086	0.224	0.958	0.198	0.855
Zn	0.433	0.350	0.790	0.860	−0.077	−0.209	0.662	0.554	0.747	0.368
Pb	0.877	0.019	0.131	0.893	−0.075	0.275	0.908	0.037	0.725	0.253
Cd	0.665	0.220	0.594	−0.020	−0.939	0.091	0.868	−0.354	0.845	−0.016
As	0.872	0.136	0.093	0.214	0.071	0.769	0.495	0.339	0.665	0.188
Hg	0.187	0.062	0.795	−0.127	−0.185	0.765	0.836	0.081	0.668	−0.144
Mn	0.140	0.476	−0.609	0.748	0.470	−0.330	0.023	0.916	−0.093	0.730
Co	0.441	0.691	−0.249	0.774	0.497	0.087	0.041	0.942	0.160	0.827
Eigenvalue	4.612	2.047	1.408	5.000	1.824	1.039	5.184	2.594	4.501	1.893
Accumulating contribution rate (%)	33.104	58.885	80.664	47.643	63.538	78.627	41.606	77.776	35.711	63.939

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
