# Peer review of "Environmental Health and Ecological Risk Assessment of Soil Heavy Metal Pollution in the Coastal Cities of Estuarine Bay—A Case Study of Hangzhou Bay, China"

_toxics, 2020, doi:10.3390/toxics8030075_

Round 1

Reviewer 1 Report

Lu S.G., Bai S.Q.: Contamination and potential mobility assessment of heavy metals in urban soils of Hangzhou, China: relationship with different land uses. Environ. Earth Sci. 60:1481-1490 (2010)

Shi G., Chen Zh.L., Xu Sh.Y., Zhang J., Wang L., Bi Ch.J., Teng J.Y.: Potentially toxic metal contamination of urban soils and roadside dust in Shanghai, China. Environ. Pollut. 156, 251-260 (2008)

To my knowledge, some papers about urban soils and road dust in the area already exist (quotations given above are not mentioned in the text). This paper covers additional elements, and comes to abbout similar conclusions.

Lines 39/40: not quite true. Agricultural areas are usually affected from Cd in phosphate fertilizers, Sb also from tires, Pb and Hg also from batteries, Pb and Cd also from paints, Zn from roofs, and Cr from vehicle abrasions. The authors should give at least an explanation, why this should be different in China.

Line 86: in the paper from 2008, Shanghai has 17 million inhabitants, and a mean temperature of 15,8°C. Are the given figures of 25 million and 17,6°C an effect of population growth and rapid global warming?

Line 113: Possibly too high results in the ICP-MS upon Cr come from Ar-C, As from Ar-Cl (to my experience), and possibly Co from Ar-F. Hopefully, the collision cell (which collision gas?) had overcome this.

Line 117: the pollution load index refers to the tolerable mlevels, which had been set by legal advices specifically for each country, basded on health ompactes. The tolerable levels set by the Shanghai administration (or maybe by central Chinese authorities), are not given, nor the analytical method.

Usually, the tolerable levels for soils are given for a grain  size < 2mm, and aqua regia digest. But within this work you had used < 0,075 mm and total digestion - leading to higher concentration values anyway, and your pollution indices may be fakes!

Line 146: The potential ecological risk index yields a contamination index with weighted eco-toxicity response, which is an interesting approach: but this eco-toxicity depends on speciation and grain size - you have to be careful, if the references refer to the same speciation (solubility) and grain size!

Table 1: for the risk model, a young man (or woman?) of 60 kg and 24 years of age has been assumed - why ? Is this the average population living in Shanghai?

Line 183: Hg had the highest variation, because it is closer to detection limits!

Table 3: due to the achievable precision, pldease shorten to 1  number after comma for Cu, Cr, Ni, Zn, Pb and Co, 0 number after comma for Mn. Because most of the datasets are non-normally distributed, a median and a range is preferable, instead of a mean and a maximum

The given data in table 3 are close to mean crust values, which is 27 mg/kg for Cu, 73 for Cr, 34 for Ni, 75 for Zn, 0,06 for Cd, 5,7 for As, and 15 mg/kg for Co. But values differ between references.

Line 352: It is not traceable, why Cr at mean crust levels should be a cancerogenic hazard. Cr is not toxic at al, just Cr(VI), which is rapidly degraded with organic carbon or microbially at this humid climate, unless you prove that chromate is stable in your soils. The higher manganese in the soils is due to the subtropical climate at your location and quite normal - do not make panics!

Have you ever cared about the geochemical environment of Shanghai? Which type of rocks, resp. fluvial and marine sediments, and which concentration ranges are expectable ? The geochemical maps refer to total digests of rocks. The soils refer to < 2mm, and aqua regia. This is different, particularly for Cr. Some data for risk assessment refer to solubilities in body fluids

Please give background values for local soils, or at least for a similar geological formation; you will not easily find a spot around Shanghai, which has not been touched by human influence

Reviewer 2 Report

The manuscript describes the evaluation of soil contamination on large area of Hangzhou bay. The authors collected 75 soil samples from localities with different land use and applied standard statistical methods for the evaluation. The manuscript was consistently prepared and the formal level high. Nevertheless, the manuscript brings some problematic parts for me that should be explained.

Materials and Methods

The authors took soil samples from topsoil (0-30cm). It should be more exactly explained. Especially, woodland soils have very different stratigraphy in comparison with agricultural soils for example. What about upper horizons of litter? Was it eliminated before soil sample taking or not? The soil sample was taken from the whole depth of 0 – 30 cm everywhere? In forest soils, the elements bound on thin layer of humic horizon usually and their concentrations in mineral horizon decreases very sharply. If you took the soil samples from 0-30 cm the intensity of elements concentration dilution was depending on the thickness of humic horizon. Was this fact treated somehow? What about the other soils?

Can you better describes the values in Environmental Quality Standard for Soils? The values are not presented in the work. Were these values derived from real element concentrations in Chinese soils or are these values recommended values only? In the calculations must be used local background value (page 4 of 20). If you do not use realistic local background value, the calculation is than only exact calculation of inexact values.

Results

Some results are surprising for me. Especially Cr that should cause the highest carcinogenic risk in this work. On the page 9 of 20 the authors present the presumption of nature source of Cr concentrations. Because measured Cr concentrations are not very high I agree with that. How can the authors explain increased Cr carcinogenic risk from natural sources (soil substrates)? It is very well known that toxicity of CR3+ form that is strongly prevailing in soils is very low. I never indicated increased carcinogenic risk of Cr with soil concentration to 200mg/kg. Opposite, Cr6+ form causes really carcinogenic risk but this form occurs in water environment or is applied on the soil by tannery sludge for example. Do have authors any explanation for that? Do not results like that lead to challenge of calculated values?

Round 2

Reviewer 1 Report

The authors were so kind to consider most of the arguments of the first run, except for note 1 in the introduction. In particular, local estimated background values have been given for Shanghai within table 3.

The reference doses RfD given in table 2, however, refer to oral ingestion of food, in particular to hexavalent chromium. The authors think that people in Shanghai feed on soil, which is obviously not true. Thus, calculation of risk assessment using these data, is doubtfull

The USEPA gives a non-effect level of > 1,5 mg/kg of Cr, see www.epa.gov/iris

Hexavalent Cr is hardly stable in soil in presence of organic carbon

Chromium in soils is not toxic at all; but there are occupational exposures from chromium vapours, or chromate containing cement, leading to cancer - which should not be mixed with expopsures from soil. Please add my statements into the respective database.

The authors should therefore delete the ecological risk assessment calculation

Reviewer 2 Report

The authors corrected the manuscript and explained some problematic aspects. I have no critical comments. 

Author Response

Thank you very much for your previous review comments.

This manuscript is a resubmission of an earlier submission. The following is a list of the peer review reports and author responses from that submission.